# Benzodiazepines: Their Use either as Essential Medicines or as Toxics Substances

**DOI:** 10.3390/toxics9020025

**Published:** 2021-02-01

**Authors:** Edilma Sanabria, Ronald Edgardo Cuenca, Miguel Ángel Esteso, Mauricio Maldonado

**Affiliations:** 1Grupo GICRIM, Programa de Investigación Criminal, Universidad Manuela Beltrán, Avenida Circunvalar No. 60-00, 111321 Bogotá, Colombia; edilma.sanabria@docentes.umb.edu.co (E.S.); ronald.cuenca@docentes.umb.edu.co (R.E.C.); 2Universidad Católica Santa Teresa de Jesús de Ávila, Calle los Canteros s/n, 05005 Ávila, Spain; mangel.esteso@ucavila.es; 3U.D. Química Física, Universidad de Alcalá, 28805 Alcalá de Henares, Spain; 4Departamento de Química, Facultad de Ciencias, Universidad Nacional de Colombia, Sede Bogotá, Cr. 30 No. 45-03, 111321 Bogotá, Colombia

**Keywords:** benzodiazepines, medical uses, toxicology

## Abstract

This review highlights the nature, characteristics, properties, pharmacological differences between different types of benzodiazepines, the mechanism of action in the central nervous system, and the degradation of benzodiazepines. In the end, the efforts to reduce the benzodiazepines’ adverse effects are shown and a reflection is made on the responsible uses of these medications.

## 1. Introduction

According to the World Health Organization (WHO), benzodiazepines are essential medicines used in the treatment of many clinical disorders, including different kinds of epileptic attacks, panic, phobias, depression, excitation, aggressiveness, anxiety, and insomnia, among others. Fast starting of action, efficiency, a minor number of collateral effects, and minimum acute toxicity are some of the advantages of using these medicines. However, these can also show serious adverse effects when they are used improperly. Some examples of this are: consumption times longer than prescribed; when they are consumed mixed with other stimulants substances like Opium; consumption of chemically modified benzodiazepines which have not been approved as medicaments; overdose; if during the treatment both driving and heavy machinery handling is not restricted; and, used for criminal purposes, when they are administered masked in drinks or food in order to steal or sexually abuse victims.

## 2. Benzodiazepines

The Benzodiazepines (BDZs) are a group of components that receive their name because in their chemical structure have a benzene ring joined to other seven-members heterocyclic ring called Diazepine (Figure 1).

There is a wide range of medicines that belong to these types of substances. The type of BDZ depends on the component present in the basic nucleus of the compound. As an example of it, Figure 2 shows several BDZ structures, in which the termination, lam, pam, etc., indicate which is its active principle.

## 3. Synthesis of Benzodiazepines

Within the BDZs group, several families of these compounds can be found differing in the position and presence of functional groups. Some of the great importance are 1,4-benzodiazepines, 1,5-benzodiazepines, 1,4-benzodiazepines-2-ones, and 1,4-benzodiazepines-3-ones. The synthesis of 1,5-benzodiazepines includes the reactions of phenylendiamines with ketones in the presence of catalysts [1,2,3,4] by condensation with α, β-unsaturated carbonyl compounds [5]. The reactions can also be carried out under solvent-free conditions [6,7] or by means of procedures under microwave conditions [8,9].

Methods for preparing 1,4-benzodiazepine derivatives include an aromatic electrophilic substitution from N-substituted anilines [10]. It is also possible via a condensation reaction between *o*-substituted benzylamines with activated aziridines by ring-opening reactions [11,12,13], among others. 1,4-Benzodiazepin-3-ones are synthesized by the condensation of 2-bromobenzylamines with amino acids [14], as well as by intramolecular nucleophylic aromatic substitution [15]; other alternate methods include cyclization process by Ugi reaction [16,17]. The synthesis of 1,4-benzodiazepin-2-ones includes the previous synthesis of the intermediate amino-ketone [18]. Moreover, cyclization processes involving an Ugi reaction as a first step [19,20]. Finally, other methods include the cascade coupling/ condensation process of 2-bromobenzylamines with amino acids [21] and Buchwald reaction [22] (Figure 3).

## 4. Benzodiazepines Action Mechanism on the Central Nervous System

Due to its high solubility in lipids, BDZ can penetrate the brain and cause rapid and effective action through interference with the nerve impulse transmission mechanism [23]. In order to understand this, it is necessary to keep in mind that the main inhibitory neurotransmitter of the central nervous system is the γ-aminobutyric acid (GABA), which has a reception place in the neurons acquiesced by a heteropentameric glycoprotein. This protein is coupled in the central nervous system, more abundant in ionic channels through which chloride ions selectively transit, they also have a site for the reception of the BDZ, known as Benzodiazepine Union Place. The BDZ bind to the binding site and modify the three-dimensional disposition of the receptor (allosteric modulation) [24,25], favoring the opening of the chloride channel by the action of GABA (Figure 4). The former process is the potentiation of the inhibitory effect of the GABA neurotransmitter so that the neuron becomes less excitable and begins to be in a neuronal inhibition state. In addition, the permanent activation of GABAA receptors causes adaptative changes that result in tolerance [26].

The GABA neurotransmitter action is decisive in the central nervous system because approximately 40% of neurons respond to its effects, so the reassuring effect is significant. The inconvenience with altering this process is that by consuming BDZ neurons get overloaded and they do not respond in the same way to the exciters neurotransmitters, and even reduces noradrenaline, serotonin, acetylcholine, and dopamine which are important in processes of wakefulness, alert, memory, coordination, in the production of glandular secretions, the cardiac rhythm control and blood pressure, among others, which in fact leads to a general sedative-hypnotic effect [27]. To avoid these inconveniences, for several decades molecules capable of being used as anxiolytics have been developed as an alternative to benzodiazepines such as bretazeni, imidazenil [28], alpidem, and ocinaplon and although in the last two cases, these were toxic to the liver, they showed that it is possible to develop anxiolytics based on GABAA receptors [29].

## 5. Pharmacological Differences between the Different Types of Benzodiazepines

Differences in action between the different types of BDZs in the brain are associated with several sub-units (α1, α2, α3, α5) present at the Benzodiazepine Union Place of the heteropentameric glycoprotein at the γ-aminobutyric acid reception site [30,31]. Although it is not yet completely elucidated, it is believed that each sub-unit regulates different actions. For example, Rosas-Gutierrez et al. in 2012 indicated that the sub-unit α1 is the most abundant and regulates the anti-convulsant, hypnotic, and sedative actions in the body, the α2 sub-unit regulates the anxiolytic effects, and the sub-units α3 and α5 are related to the relaxing muscle effect. These authors also indicate that, to a greater or lesser degree, each BDZ joins to an α sub-unit. Thus, classic BDZs like Diazepam get joined to the α1, α2, α3, α5 sub-units, while other medicines such Zoldipem has a specific affinity for the α1 sub-unit and barely any affinity for the α5 sub-unit [25]. On the other hand, Engin et al. indicate that positive allosteric modulators of α5 sub-units might be useful to attenuate interference-related cognitive symptoms and hippocampal hyperactivity in some psychiatric disorders [31]. While Zhiqiang indicates a possible role of α3 in the abuse potential and sedative effects of benzodiazepine-type drugs according to studies carried out in rhesus monkeys [32].

Regarding the pharmacological sphere, the BDZ action can vary slightly depending on its nature [33] and its selectivity with respect to the receptors. These differences are mainly manifested in the hypnotic/anxiolytic sedative effect, in the time of life, in the administered doses [34], in the elimination velocity [27], in the hepatic metabolism [35], in the number of doses (single or multiple), in the drug liposolubility, in the administration mechanism (oral, intramuscular), in the potency, in the action onset (fast <1 h, medium 1–2 h, slow >2 h) [34], and in the accumulation and myorelaxing effects, among others [36]. As an example of this, we can cite BDZs with different half-life times; among them, it is possible to find Bentazepam, Midazolam, and Loprazolam, with a half-life time of less than six hours; Alprazolam, Bromazepam, Nitrazepam, with a half-life time between 6 and 24 h; and Diazepam, Ketazolam, and Flurazepam, with a half-life time greater than 24 h [37].

## 6. Degradation of Benzodiazepines in the Organism

BDZs are metabolized in the liver with the intervention of microsomal enzyme systems. The first stage of the process (Figure 5) is the hydrolysis of the Diazepine ring at positions 2 or 5 in order to obtain the intermediates 1 and 2; afterward, a hydrolysis process takes place to produce 2-aminobenzofenona and glycine, which are excreted through the urine.

## 7. Toxicological Effects of Inappropriate Use of Benzodiazepines

### 7.1. An Essential Medicine That May Become a Weapon for the Crime

Medicaments arise to save lives, to provide solutions to sick people, to facilitate their treatment, and to make easier the course of many sufferings. Besides, medicaments save enormous amounts of money because they avoid long hospital stays and large and prolonged disabilities. Particularly during the covid-19 pandemic, the change in the family organization, of working, and the domiciliary confinement has induced feelings of helplessness, abandonment, loneliness, and insomnia among others [38]; therefore, prescription for anxiolytics medications increased 34.1% from mid-February to mid-March and it has become an option to help stabilize the mental health of the population [39]. However, regardless of these advantages, medicaments require responsibility in their administration and management, like following the medical guidelines regarding administration and dosage, the lapses of administration and the total time of treatment, as well as their use only for therapeutic reasons. Because if they are not used properly, they can cause serious problems of addiction or they can even become real delinquency weapons like it will be explained below for the case of BDZs.

BDZs started to be used in 1957 as a safer alternative to the use of barbiturates, which in turn replaced all the opiate derivatives when they were prohibited as sedatives [40]. The first BDZ introduced in the world was chlordiazepoxide (Figure 6) in 1960 [41], and from then on have appeared a wide range of types of different BDZs [40]. BDZs have been used in the treatment against insomnia, agitation, anxiety, and convulsions [37]. According to the 21-list published by the World Health Organization in June 2019 [42], they are considered as essential medicaments. 

However, due to the fact that they are depressor agents of the central nervous system, they are also used for criminal purposes. In some cases, these substances have been used to perpetrate crimes, by administering them to the victims in drinks or foods [43], with the added disadvantage of the difficulty of their detection, as a consequence of both their rapid metabolization process and that their concentration varies according to the consumption of liquids made by the victim [44], which make their identification and range of affectation very difficult. Thus, these substances are often used to perpetrate rapes; for which a non-suspicious person administers it to the victim in her drink and subsequently rapes the victim. People who have lived that type of experience usually manifest waking up in unknown places, being improperly dressed, and not remembering anything related to the assault [45]. They have also been used by serial killers; the most famous case was the one known as “oral hygiene”, in which four nurses killed near to 43 persons, mainly the elderly, between 1983 and 1987, through the administration of BDZs or insulin in the mouthwash, which kills the victim in a maximum of one hour. This happened in Lainz Hospital, in Vienna, Austria. The nurses argued for their defense, compassion for the sick person, while the police hold that the facts had to do more with a kind of annoying people elimination plan [46] (Figure 7).

Another case with the same BDZ (flunitrazepam) was published in 2013; the victims were intoxicated and robbed at the railway station. This conduct was repeated every Friday by Japanese citizens between January and August 2003. They offered welcome cookies to Asian travelers. Twelve of the 16 victims tested positive the test of flunitrazepam. Finally, the police send undercover agents and managed to capture those responsible. This case shows as a mishandled drug can facilitate the criminals’ activities [47].

### 7.2. Benzodiazepines Effects on Aggressors

Sometimes, criminals commit their crimes under the influence of BDZs, consumed either alone or mixed with other stimulants substances like cannabis or methamphetamines, which act on the central nervous system. Under the effects of these substances, criminals can commit the most horrifying crimes against victims due to disinhibition effect, invincibility, and estrange aggressiveness feelings produced by the drugs [38]. A relatively recent case was known as the Stephen Paddock one, in the United States. This guy shot and killed 58 persons and wounded more than 500 during a concert in Las Vegas, the first of October 2017. This case generated a huge debate because it is believed that he perpetrated the attack under the effects of Diazepam Figure 8, drug that had been prescribed to him some months before [48].

It is known that these criminals usually do not think clearly under the effects of the drugs, and after the attacks can have integrated amnesia episodes [49]. In fact, the long-term consumers of these substances can suffer incapacity to feel pain or pleasure, an alteration known as emotive anesthesia. A study intended to evidence the magnitude of this problem was carried out in France in 2017; it was corroborated that 68 of 140 criminals with any psychiatric disorder done their crimes under the effects of BDZs [50].

The consumption of BDZs mixed with other stimulants substances, like Opium, is sometimes a highly controversial legal practice, supported by a medical prescription, which generates a great ethical–scientific debate. In the United States, it was deeply revised between 1999 and 2013 because during this period deaths for opioids increased by 819%. As a result, it was established at the international level some policies directed to both supervise some BDZs and the public sensitization about the adverse consequences of their consumption mixed with other stimulants substances of the central nervous system. Moreover, some mechanisms were also created to make aware the population about the consequences of consuming these medicaments falsified, adulterated, or changed for others with a different medical effect [48].

### 7.3. Suicides Mediated by Benzodiazepines

According to a study carried out by Berecz et al. [51], in Hungary, a country with a population similar to Austria, Honduras, or Sweden, the number of suicides related to chemical substances was 742 during the last decade of the last century, 1990 to 2001. Of these, 179 were related to psychoactive agents such as antidepressants, selective inhibitors of the serotonin reception, barbiturates, and BDZs, with the latter three being the main causes of suicide, 79% of the total, and overdose the main method for carry out the suicide.

These data highlight a major problem caused by the inappropriate use of medications. There are studies evidencing that it is possible to reduce the number of suicides by implementing opportune treatments to fight mental disorders and depression. This must be linked with improving safety in the pharmacological system of people at risk of suicide, as well as the implementation of strict access to these medications [51].

### 7.4. Benzodiazepines and Traffic Accidents

BDZs, by presenting a sedative effect, alter the psychomotor efficiency and can therefore cause traffic accidents. This is due to the fact that these medicines increase the reaction time and, therefore, cause motor disorders [35]. According to a study published by Roth and Roehrs [52], psychomotor disturbances are reflected, among others, in increased reaction times, attention deficits, and response errors. These effects are directly linked with the amount of substance consumed, the type of BDZ, its residual effect, and the time it takes to start driving after the ingestion [52]. In general, those who are under treatment with BDZs are recommended to pay special attention to these undesirable effects and to avoid, as far as possible, the driving of vehicles, as well as the use of heavy machinery during the treatment.

### 7.5. Adverse Effects of the Benzodiazepines Consumption by the Elderly

It is known that people’s metabolism decreases with age, leading to the appearance of neurodegenerative and metabolic diseases [53]. The aging of the organism is reflected in the less effective way to metabolizing medications so that in the elderly, the effect of BDZs is longer and the drug gets accumulated in the system. In neuronal terms, the elderly have fewer neurons, so the dose prescribed for them is usually less than that of the young [27]. The issue is so serious that these medications are included in the Beers list that brings together restricted medicines for the elderly [54].

The adverse effects that the elderly can experience from the use of these medications are confusion, night ambulation, amnesia, loss of balance, somnolence, among others. Excessive sedation can cause them somnolence, lack of concentration and of coordination, muscle weakness and dizziness, effects that can cause falls, fractures, and traffic accidents [27].

BDZs are currently widely used in the elderly as a treatment for anxiety and sleep disorders, although under a restricted medical prescription. As a general trend, it is recommended that their use does not exceed three to four weeks. Thus, the Medicines Spanish Agency points out that the use of BDZs as a treatment for insomnia should not exceed one month; however, in the case of anxiety treatments, it extends this time, although pointing out that it should not exceed three months. In conclusion, it is necessary to pay more attention to the dose used and the time of these medicaments are used because they can generate dependence and tolerance [54].

### 7.6. Adverse Effects on Pregnant Women and Newborn Children

During pregnancy, BDZs permeate the placenta and reach the fetus, where they undergo slow metabolization, causing the medication to accumulate and causing adverse effects that are evident after birth. These effects may include delay in the growing process, sedation, inability to breastfeed, abstinence symptoms, and also the newborn may suffer extreme excitability and feeding difficulties. In the long term, it is feared that they may even suffer from mental retardation, hyperactivity, and other disorders derived from delayed brain development [27]. Besides, it has been corroborated that children exposed to BDZs during pregnancy are prone to hospitalization in intensive care and also that they tend to have a smaller head circumference than non-exposed children. Because of this, it is highly recommended to avoid the use of BDZs during pregnancy, and if it is strictly necessary to treat anxiety, it is advisable to use other medicines that do not represent any risk [55].

## 8. Chemically Modified Benzodiazepines

To the aforementioned problems, we have to add that it is currently possible to find associated drugs known as “designed benzodiazepines” on the market, belonging to the new group of drugs called “new psychoactive substances” (NSP). These substances, which are obtained by modifying the basic BDZs (Figure 9), have not been approved as medicines, so they could be dangerous for the health of consumers [56,57].

As an example, in late 2017, the case of a girl who was stolen after drinking tea with a man she met via the Internet and who added flubromazolam in her drink was reported. The victim, a 21-year-old, was confused for 1 h, then remained unconscious for another two more hours, and finally woke up in a hotel room without her wallet and cell phone. She reported the case to the public security authority and the suspect was arrested a few days later. Police found two bottles of a transparent liquid in his home. The liquid is known as Lie Yan and contained a designed BDZ called flubromazolam that is considered illegal in Europe since 2014. Flubromazolam has a strong and prolonged depressive effect on the central nervous system that, when the poisoning is severe, can produce serious complications such as hypoxial ischemic, respiratory failure, hypotension, and finally brain damage [58].

The problem gets worst if we consider that many of the commercialized BDZs may actually be counterfeits of others of the same species but with different pharmacological activity, or may even contain synthetic opioids such as U-47700 or other central nervous system stimulants. As an example, in January 2017, in the United States, the director of a virtual pharmacy was sentenced to eight years in prison for distributing Flubromazolam tablets instead of Alprazolam. Similar cases of replacement of medicine by other of the same composition family have been reported in Finland, Singapore, and Malaysia, among others [59].

## 9. Inadequate Use of Medicaments (Benzodiazepines) in Numbers

According to a study carried out by the United Nation Office on Drug and Crime (UNODC), between 1999 and 2013, in the United States, the mortality rate in which the use of BDZs was involved increased by 514%, while the number of deaths caused by the use of BDZs mixed with opioids increased to 175,000 deaths. The BDZs most associated with deaths were Clonazepam, Flunitrazepam, Oxazepam, Alprazolam, and Diazepam, with the last two substances having the highest mortality rates between 2010 and 2014 [59]. Figure 10 shows the structure of these compounds. A study carried out in 2019 by Baillargeon et al. in older adults with chronic obstructive pulmonary disease revealed that the use of opioid and benzodiazepine alone or in combination were associated with increased adverse respiratory events [60].

In Europe, BDZs are the prescribed medications most related to cases of acute intoxication. Between 2013 and 2014 there were registered 320 cases of hospitalization to patients for the consumption of cannabis mixed with any other type of drug, including BDZs, which represented 22.8% of all cases, just behind the alcohol poisonings, that represented 59.4% of cases. In 2015, in Scotland, 706 deaths related to the use of drugs were reported, among which 57% were related to the use of BDZs, finally, 57 cases were confirmed for these medicines. The first international attempts to control the consumption of BDZs took place in 1984. These attempts were intended to control the incorrect consumption of these drugs but also to guarantee the use of those considered essential for human health. Some of the countries that currently monitor these medicines are the United Kingdom, Denmark, Finland, Switzerland, Sweden, Turkey, Korea, and the United Arab Emirates [59].

An option to control the consumption of BDZs and its dependence in patients with insomnia problems is the ramelteon. A study carried out by Naono-Nagatomo et al. indicates that additional administration of ramelteon can reduce the benzodiazepine used, improve clinical symptoms, and it does not produce side effects or dependency [61].

## 10. Interference with Other Medications

As it was previously mentioned, BDZs are depressant of the central nervous system and, therefore, can have several interactions with other medicines that work on the system. Thus, for example, with barbiturates and alcohol, BDZs can prolong the sedative effect [27], while their presence inhibits the effect of Levodopa, which is a medicine used to treat Parkinson [62]. Valproic acid, estrogen, heparin, disulfiram, digoxin, erythromycin, antacids, diltiazem, and phenytoin are medications that can enhance the action of BDZs if administered simultaneously [34].

## 11. The Other Side of the Benzodiazepines

Until now, the dark side of BZD has been presented and, in light of this, everything seems to indicate that they should not be prescribed. However, at this point, it is important to ask why then they are considered essential medicals for the World Health Organization (WHO), and also if it is justifiable to deprive people who need them of these medicines to prevent their improper use [63]. Below, it will be shown how BDZs came about and why they are currently considered essential medicines. Its correct use and the benefits they have in medical applications will be explained.

### 11.1. Medical Use of Benzodiazepines

BZD are the most prescribed psychotropic drugs because they have a wide therapeutic range [35]. They are mainly used for their sedative–hypnotic effects, as anticonvulsants, anxiolytics, muscle relaxants; in addition, they are used to treat insomnia and problems derived from the suppression of alcohol consumption (alcohol withdrawal syndrome) [36]. In cases of panic or anxiety crisis, they are also highly recommended [64]. They are also used as antiepileptic, as long as health personnel are careful administrating small amounts of them at the beginning of the treatment because, otherwise, they can generate tolerance [34]. Finally, BDZ are the favorite medications to anesthetize patients in odonatological surgeries, since they decrease the tension and anxiety caused by the intervention [40]. In some circumstances, BDZ are used to treat eclampsia and preeclampsia [65], and some types of 1,5-BDZ may have anti-neuroinflammatory activity [66]. Most of the uses mentioned above are derived from the BDZ depressant effect on the central nervous system, and that is why it makes them appreciable as psychotropic drugs. Table 1 shows the medical uses of BDZs, their advantages, and the names of other medicines that are used for the same purpose.

### 11.2. Who Prescribes Benzodiazepines

BDZs are usually prescribed by general doctors. According to a study carried out in Valencia, Spain, on 413 patients, 78.9% of them manifested that this medication had been prescribed by a primary care physician, 5.2% said that the medication had been prescribed by a psychiatrist, and the remaining 15.9% by a specialist [54].

Nevertheless, many people continue using these medications even after the prescription time has ended; patients continue using them for a long period of time because they become accustomed to its effects to carry out their daily routines, they experience anxiety and subsequently, begin to suffer withdrawal symptoms if they do not consume them. For example, in a study carried out in Spain in 2012 with 314 patients, 67% used benzodiazepines for more than one year, and only 5% of patients with insomnia knew that the duration of treatment should be less than one month [54]. Patients often return to the doctor to update the prescriptions or even go to other doctors or hospitals to increase the dose or start with a new treatment.

BDZ can be used for recreational purposes because mixed with other drugs, can increase the degree of excitement [40]. The problem has reached such levels that in some countries medications such as Diazepam and Alprazolam are prescribed for recreational purposes and are generally used in higher doses than prescribed.

BDZ are also used to relieve the withdrawal symptoms caused by other drugs like Opiates, Cocaine, Amphetamines, and Alcohol. The means of obtaining them range from falsified prescriptions, pharmacy robberies, and illegal imports. The benefits of these pharmaceutical compounds will be described in the next paragraph.

### 11.3. Benzodiazepines Benefits

BZD have many advantages over other substances. Among them, it is possible to find the quick action starting and its optimal tolerance at the beginning of the treatment [80]. Their molecular action mechanism on the central nervous system is widely known, and they are safer and have less dependency risk compared to barbiturates. Regarding insomnia, BDZs have a similar effect, causing a deep and restful sleep feeling, and they also have constant action, reducing the times when the patient wakes up, increasing the resting time [40].

For the treatment of anxiety and depression, BDZs are the preferred medicines in the pharmacological sphere due to their fast stating of action, efficiency, the least number of collateral effects, and the minimal acute toxicity compared to other medications [102]. BZDs are usually helpful in saving the patient’s life in a wide range of clinical disorders such as panic, phobias, or different types of epileptic seizures or those caused for drug overdoses. BZDs have also been shown to be highly effective in treating alcohol withdrawal symptoms, to be used as anesthesia, and also as adjuvants in the treatment of muscle spasms [35].

Besides, BDZs have the advantage that, in the absence of other inhibitors of the central nervous system, they do not cause respiratory or cardiovascular deficiencies that can cause death [37]. Therefore, it is possible to say, based on what has been said, that the adverse effects of BDZs are a consequence of its improper use, usually because its use for a longer period of times than the prescribed, but if they are used under prescribed recommendation, they are usually an effective tool to treat all the disorders mentioned above.

### 11.4. How and Why Did Benzodiazepines Start to Be Used

In ancient times, several disorders such as depression, nervousness, aggressiveness, anxiety, and insomnia were treated with a wide variety of chemical substances derived from natural products. Egyptians used Cannabis and Opium, whose active principle is Morphine, although they have other alkaloids. On the other hand, the Hebrews and the Babylonians used the Mandrake, whose active principle is Atropine, an alkaloid that competes with the neurotransmitter Acetylcholine for a binding site in the neurons but does not activate the receptor. The Orientals and Greeks used henbane and jimson, which are plants that contain tropics alkaloids such as Scopolamine, Hyoscyamine, and Atropine [103].

In the 19th century, things changed when some new substances were synthesized and some active principles, such as Morphine and Hyoscine, were isolated from Opium, Mandrake, and the Seaweed, becoming new medicines for the treatment of these disorders [103]. In 1864, Von Baeyer synthesized Barbituric Acid, a precursor of the Barbiturates, which were used as sedatives before the second half of the 20th century, when these substances replaced opiates [40]. These medicines were effective in treating anxiety and several types of seizures. They were also used as hypnotics in order to cause mild sedation or total anesthesia, but they generated physical dependence, withdrawal symptoms, and safety problems that generated high mortality rates, which is why their use began to be restricted. BZDs, then, emerged as an effective means of treating diseases like insomnia and anxiety [104]. BDZ came into use for the first time in 1960 with the introduction of Chlordiazepoxide and then Diazepam emerged as one of the most known BDZs [23]. Currently, within the family of BZDs, there is a wide variety of them that differ in their forms of administration, dose, selectivity, onset of action, duration, and potency, among others [104].

## 12. Some Efforts that Aim toward Obtaining Benzodiazepines with Less Adverse Effects in the Long-Term

For many years, huge efforts have been made in order to develop new types of BDZs that avoid excessive drug consumption. Authors such as Rosas-Gutierrez have looked at another possible beneficial effect by directing efforts towards obtaining BZDs, which are specific for the α2 sub-unit, that controls the anxiolytic effect, but trying not to affect the sedation and anesthesia [105].

Regarding the effect of the chemical structure of these phycopharmacos, it is known that there are currently new studies that aim to create new, more specific, BZDs with low toxicity and better ranges of tolerance. In this sense, some studies have found that the 1,5-BDZ have fewer adverse effects than 1,4-BDZ [103]. On the other hand, in 2010, Ben-Cherif et al. showed that BDZ activity depends on the existence of an electroattractive group in position 7 [105]. For example, chloride is determinant in the anticonvulsant activity and in the activity related to the sleep in the 1,5-BDZs, while the NO2 group is related to the hypnotic activity [106].

It is important to remember that all the substances could be toxic if they are not used properly; even water could be lethal when reaching the lungs or when it is drunk in huge quantities. In the case of drugs, it is necessary to remember that they have clinical effects and hence they are used to treat many diseases; but if they are not used properly, without taking into consideration their possible effects on human health, they can be turned into dangerous threats to health or dangerous arms. Particularly, in the case of BZD, it is fundamental to strictly follow the recommendation given by the professional that prescribed them. These medications should not be administered to pregnant or lactating women; they must not be used mixed with other medicines or drugs; it is also recommended that they are used in their minimum effective doses, and the time of use must be limited depending on the medical recommendations [54]; besides, they should not be administered to patients with respiratory problems; when used in the elderly, they have to be of quick elimination-type [40], and once the treatment is over, these medications must be gradually withdrawn, because some of them may cause withdrawal syndrome [59]. In cases where BDZ is not possible to be used, it is recommended to use selective serotonin reuptake inhibitors (ISRS) or non-pharmacological therapies behavioral-cognitive-type [35]. Finally, it is important to remark the fact that the various dangers associated with BDZs are linked to the use of these medications by inexperienced persons, who prolong the treatment without medical permission, who use them for recreational purposes while ignoring their consequences, or who use them consciously to sedate and attack their victims. Without prevention, the use of any medicine, even Acetaminophen, could be dangerous and toxic, making it mandatory to control the acquisition of these medications internationally, to make aware the world population about the risk and possible consequences of their inadequate use, and in the case of BZDs, to treat mental disorders on time.

## 13. Conclusions

Nature, characteristics, mechanism of action, and easy degradation in the organism are properties of BDZs that make them useful in medicals applications for the treatment of epileptic attacks, panic, phobias, depression, excitement, aggressiveness, anxiety, and insomnia, among others. Therefore, it is no coincidence that the World Health Organization (WHO) considers them essential medications. Nevertheless, it is required to use them responsibly, increase their regulation worldwide, and establish responsible consumption programs to prevent that people self-prescribe, increase treatment times without medical permission, use them for recreational purposes or mix with other stimulant substances like cannabis, methamphetamines, or alcohol, among others. In addition, it is necessary to prevent the consumption of designed BDZs, their interference with other medications, the risk of suicide, their use when driving or operating heavy machinery, adverse effects in the elderly, pregnant women, and newborn children, or even to prevent their use for criminal purposes, but also to ensure that these essential medicines are available on time for all persons who require treatment.

## Figures and Tables

**Figure 1 toxics-09-00025-f001:**
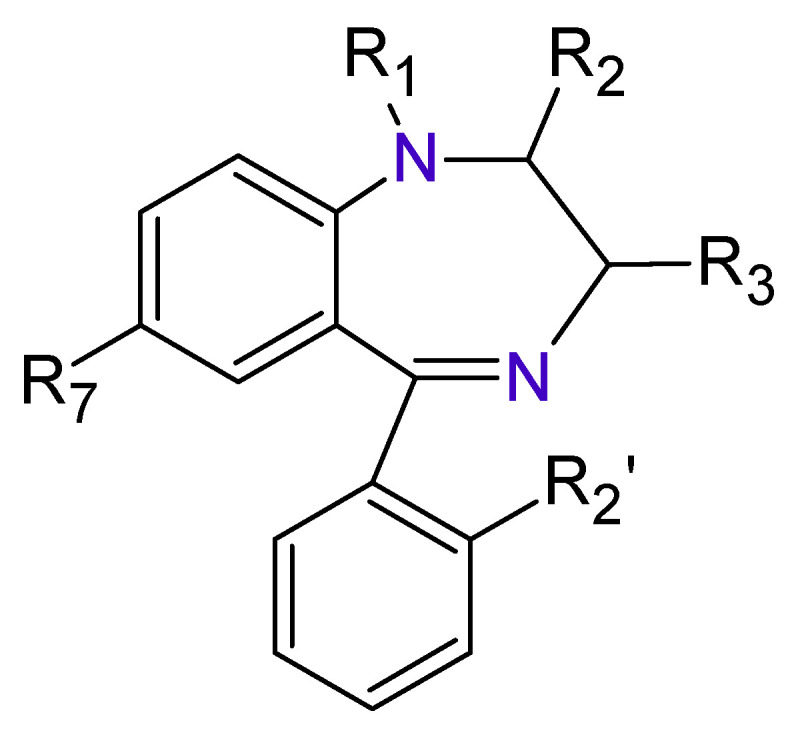
Benzodiazepines basic structure.

**Figure 2 toxics-09-00025-f002:**
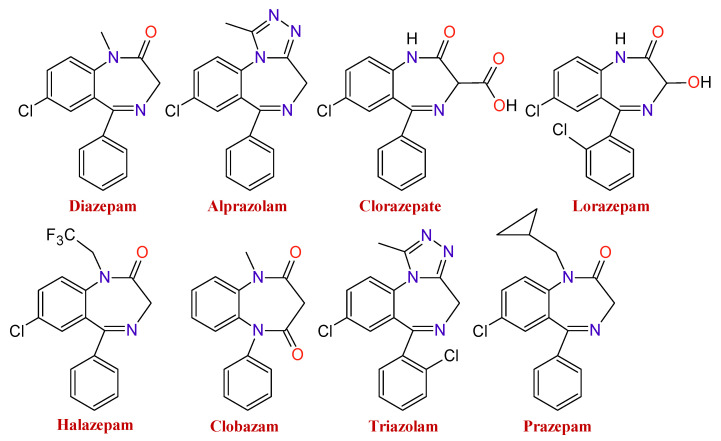
Chemical structures of some benzodiazepines.

**Figure 3 toxics-09-00025-f003:**
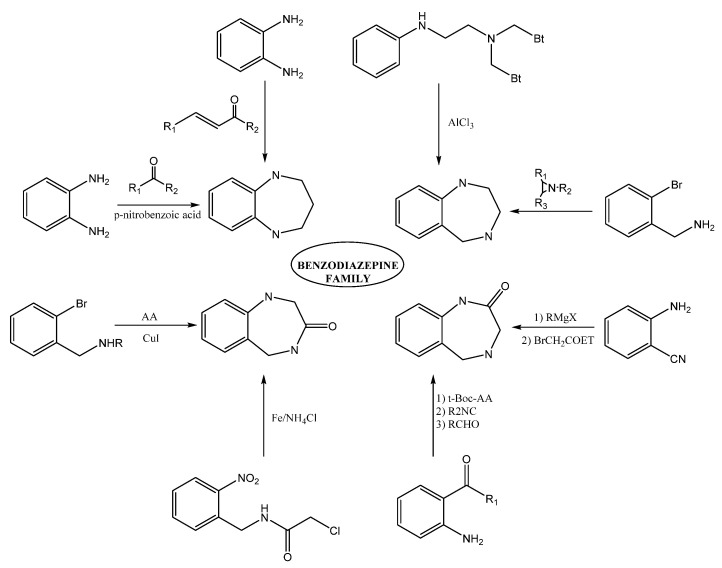
Some synthesis routes toward benzodiazepines.

**Figure 4 toxics-09-00025-f004:**
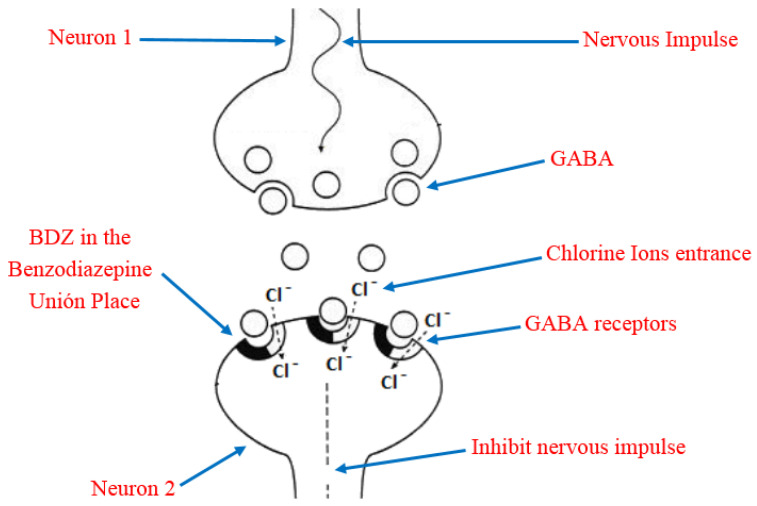
Representation of the benzodiazepines (BDZ) action mechanism and their role in the nervous impulse inhibition.

**Figure 5 toxics-09-00025-f005:**
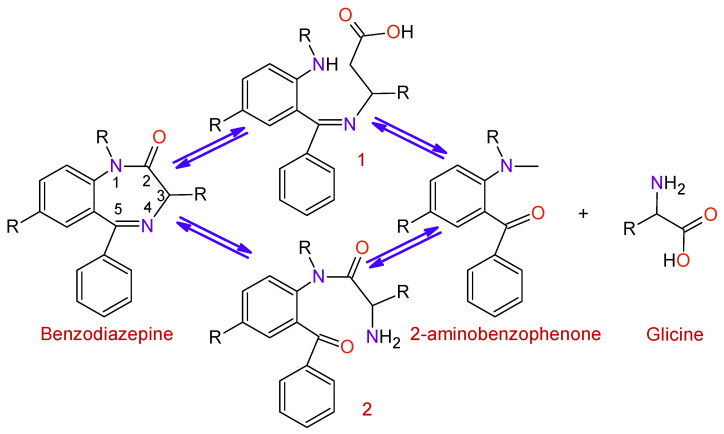
Degradation of benzodiazepines into 2-aminobenzophenone and glycine.

**Figure 6 toxics-09-00025-f006:**
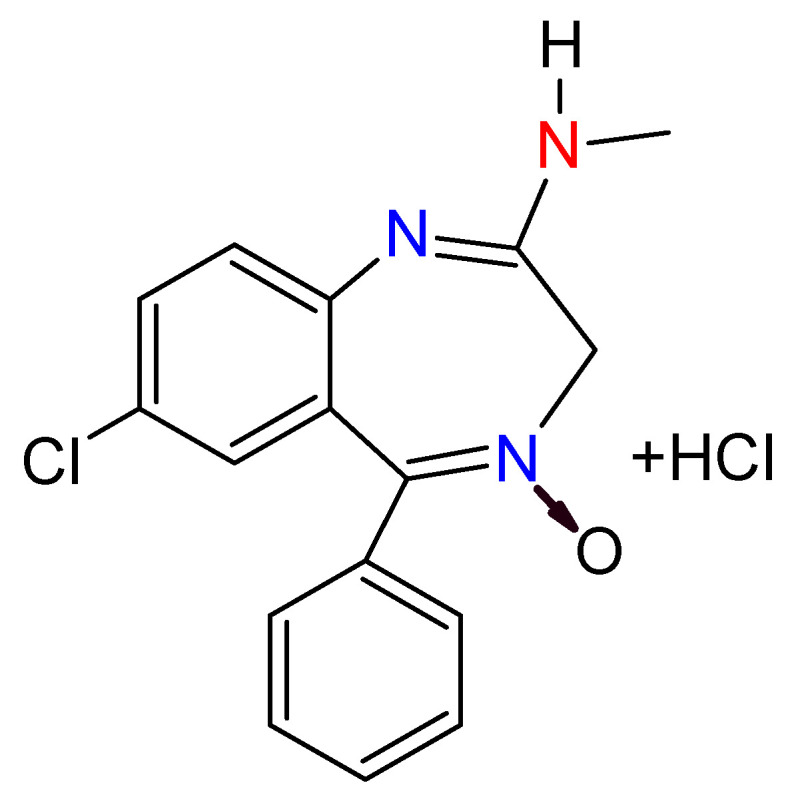
Chlordiazepoxide structure; it was the first benzodiazepine introduced in the world.

**Figure 7 toxics-09-00025-f007:**
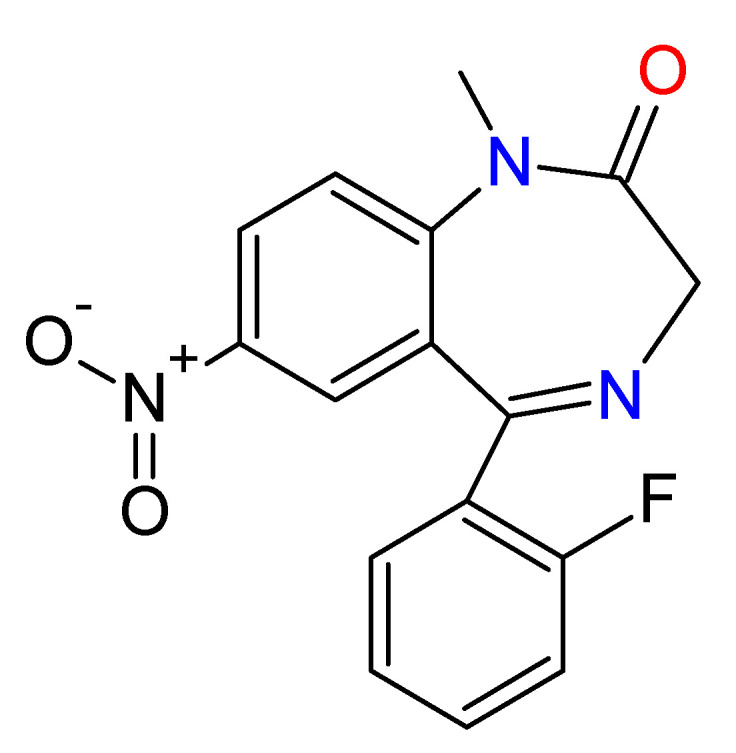
Chemical structure of the benzodiazepine (Flunitrazepam) that, together with insulin, was administered into the victim’s mouthwash by the nurses Maria Gruber, Irene Leidolf, Stephanija Meyer, and Waltraud Wagner (the “*Lainz Angels of Death*”), murdering 43 persons in Vienna, Austria.

**Figure 8 toxics-09-00025-f008:**
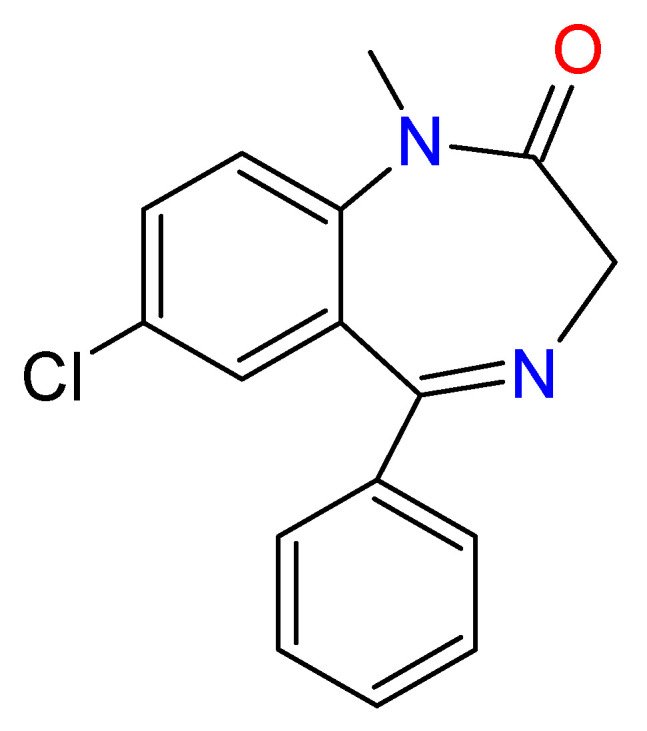
Chemical structure of Diazepam. It is believed that, under its effects, Stephen Paddock killed 58 persons during a concert in Las Vegas (USA), the first of October 2017.

**Figure 9 toxics-09-00025-f009:**
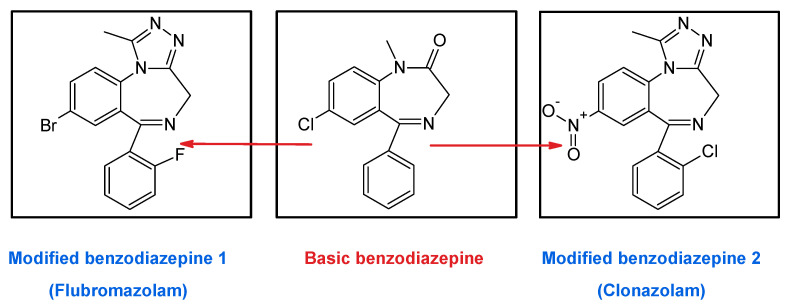
Representation of the modification of basic benzodiazepines to obtain new types of benzodiazepines.

**Figure 10 toxics-09-00025-f010:**
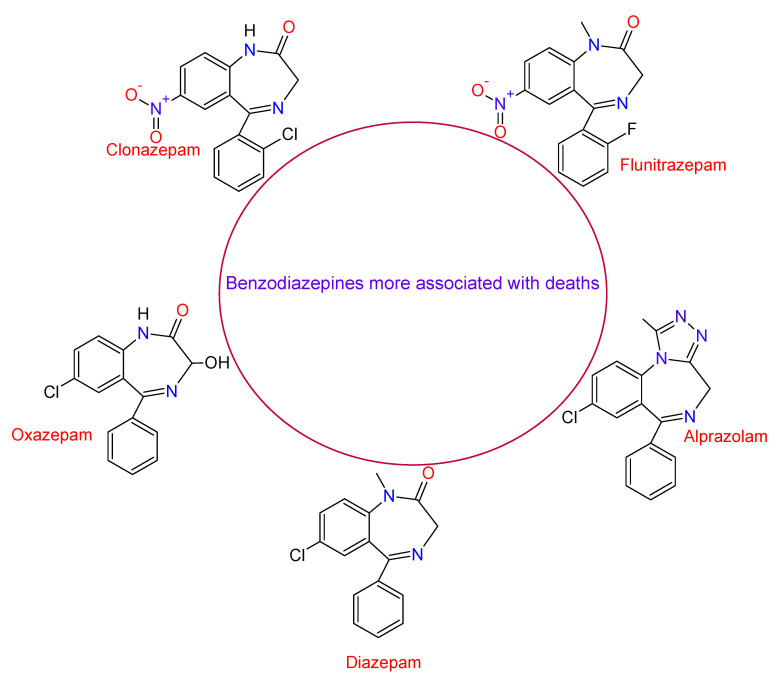
Structure of the benzodiazepines more linked to deaths.

**Table 1 toxics-09-00025-t001:** Medical uses of benzodiazepines and the comparison with other medicines used for the same purpose.

Benzodiazepine Name	Some Medical Use	Other Medicines Used for the Same Purpose	Advantages and Disadvantages of Benzodiazepines
Diazepam	Anxiety disorders, epilepsy, alcohol withdrawal symptoms (AWS), muscle spasms [67]	Fluoxetina (Prozac, Sarafem, Fontex), Escitalopram (Lexapro), Paroxetina (Paxil) (selective serotonin reuptake inhibitors) [67]	The clinical efficacy of Diazepam is recognized because it rapidly reaches its potency of action, by its safety, and high bioavailability. Among its disadvantages is that it can cause urinary retention and compartment syndrome [68]
Lorazepam	Treatment of alcohol withdrawal symptoms (AWS) [69], treatment of catatonia [70]	Benzodiazepines are the standard although other alternative agents as phenobarbital also are used frequently for the treatment of AWS [71,72], whereas for catatonia, electroconvulsive therapy is usually used [73]	Lorazepam is attractive for treatment of AWS because it is likely to accumulate in liver less than other substances [69]. For the treatment of catatonia, Lorazepam is considered safe and effective (recovery rates >80%). Independent of the blood concentration, the lorazepam can cause significant impairment of driving and psychomotor abilities [68]
Alprazolam	Treatment of panic disorder (PD), agoraphobia [74], and depression [75]. In addition, it is used as an anxiolytic agent	For PD, fluoxetine and escitalopram (selective serotonin reuptake inhibitors) are also used [76]. For agoraphobia, the treatment with paroxetine, sertraline, citalopram, escitalopram, and clomipramine showed good results [77]	Alprazolam is a high potency BDZ and, due to its effectiveness, it is the most widely used in the pharmacological treatment of panic disorders [76]. It has a recognized efficacy for the treatment of agoraphobia [78]. Its disadvantages include sedation, withdrawal symptoms, and abuse [79]
Clobazam	It is used as anxiolytic, anticonvulsant and antiepileptic agent [80]	In the treatment of seizures valproic acid, lamotrigine, and topiramate, among others, are used [81]	Clobazam has shown great efficacy and high safety in the treatment of refractory epilepsy in both children and adults; in general, it has shown great benefits for epileptic patients [82]. Moreover, Clobazam (1,5-benzodiazepine) exhibits a better pharmacological profile in both short and long-term seizures treatments than 1,4-benzodiazepine [83]. One case of photo-induced toxic epidermal necrolysis caused by clobazam was reported [84]
Clonazepam	Treatment of schistosomiasis, a disease considered to be a serious public health problem [85]	Praziquantel can also be used to treat schistosomiasis, although its use can cause resistance to parasites [85]	Methyl-clonazepam has a schistosomicidal effect in humans and could be an alternative to Praziquantel [85], but it can cause adverse effects on the central nervous systems derived from differences in Clonazepam metabolism [86]. Among the adverse effects is that it can cause drowsiness [87]
Bromazepam	Bromazepam is a BDZ used clinically as anxiolytic [88,89,90]	Dexmedetomidine is a pharmacological option as an anxiolytic to bromazepam [91]	Studies on psychiatric patients suggest that it is better as anxiolytic than diazepam and lorazepam [88]. In addition, it does not produce negative effects on cognitive and motor response [92,93]. Its main point again is that it produces dependency [94]
Midazolam	Midazolam is used for the treatment of the epileptic state [95] and as a sedative [96]	For the treatment of epilepsy, Sodium Valproate, Cannabidiol, Ethosuximide, Lamotrigine, among others, can also be used [97]	The use of Midazolam for the treatment of epilepsy is associated with a shorter seizure duration [95]. Some adverse effects of intranasal medication are nasal burning and bitter taste [98]
Flunitrazepam	The use of Flunitrazepam is licensed in Europe, Asia, and Latin America for the treatment of insomnia, as pre-anesthetic and as sedative-hypnotic agent. Nevertheless, its use has not been approved in the U.S. [98]	Other drugs used as hypnotic-sedatives are Barbiturates, Diphenhydramine and receptor agonists such as Zopiclone, Zolpidem, Zaleplon, and Eszopiclone [99]	Flunitrazepam has no advantages over other BDZs. However, it is very popular among alcohol and drug abusers and it may produce violent behaviors [100] including being implicated in sexual assault [101]

Caution: The information given in this table is not intended to recommend the use of any benzodiazepine, local legal situation and best medical practice result in prescription practice.

## Data Availability

Not applicable.

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
