# Peer review of "Benzodiazepines: Their Use either as Essential Medicines or as Toxics Substances"

_toxics, 2021, doi:10.3390/toxics9020025_

Round 1

Reviewer 1 Report

Previous suggestions were not fully implemented. Terms that do not exist in English, for example "Benzodiazpeine Union place" are still used. 

I think the English should be improved, and otherwise the paper is interesting. 

Reviewer 2 Report

The authors have adequately addressed all my concerns. 

Reviewer 3 Report

I found the paper to be interesting and well-written. It's clear in its purpose, discussed nicely and complete. 

This manuscript is a resubmission of an earlier submission. The following is a list of the peer review reports and author responses from that submission.

Round 1

Reviewer 1 Report

I attached a word file in which I made my comments.

Reviewer 2 Report

Sanabria et al. present a review on benzodiazepines. The broad scope which even extends to use of benzodiazepines for criminal purposes is very informative, the compiled material is of broad interest.

However, prior to publication, some issues have to be adressed as indicated below:

Critical points:

The English is in part very poor, the work urgently requires language editing.  Some examples: The benzodiazepine binding site is called „benzodiazepinic union place“; modified or derivatied is called „modificated“ (Legend to Figure 9); chloride ions should not be called „chlorine“, glycine is sometimes spelled correctly, somethines as „glicine“; several sentences (e.g. p3, lines 71ff) are incomprehensible; persond are kidnapped and not „stolen“, …..   

Chapters 4 and 5 (action mechanism and pharmacological differences) are based in part on outdated literature, and the total number of citations is too low to correctly reflect the complexity of the topic.

The role of individual alpha isoforms in behavioral effects is much more controversial, see (and cite) for example Phil Skolnick Trends Pharmacol Sci. 2012 Nov;33(11):611-20. doi: 10.1016/j.tips.2012.08.003.

IUPAC conform naming of receptors and subunits should be used. Isoform number should not be subscripted (e.g. p4 line 93ff)

Pp4/5: The information on half lives is misleading because long lived metabolites e.g. of ketazolam are ignored

P5, lines 125ff, the historical development is misleading and inaccurate – chlordiazepoxide was released only on 1960, the years before went into the time between discovery and approval.

Figure 9: The authors write about modified benzodiazepines, but the figures display approved drugs (halazepam, prazepam) – genuine „new benzodiazepines“ should be displayed instead

P9, line 264 „study carried out by the [48]“ is incomplete, carried out by whom or what??

Table 1 could be useful, but should be modified to better reflect common uses of the benzodiazepines, and also to give a more balanced view of alternatives: 1) Not only lorazepam, but many other benzodiazepines are used to treat AWS, while baclofen is not standard;  2) the description of bromazepam is very diffuse: „psychic function, behavior or experience“ ??? The main indication for bromazepam is simply anxiolysis. Its main downside is strong physical dependence. A more balanced discussion would be needed, or bromazepam should be deleted from the table to not give the misleading idea that is has a favorable profile. 3) Generally, table 1 should present only benzodiazepines with clear advantages (i.e not feature flunitrazepam, which is well discussed as a heavily abused benzodiazepine). Or alternatively, the table could be divided into „more recommended“ and „more problematic“ benzodiazepines

Ethical concerns: There is an ethically thin line between attesting a favourable profile to a drug, and „advertising“. Table 1 information should be accompanied by a statement which indicates that local legal situation and best medical practise result in prescription practice, and that a review is not intended to recommend prescription or use of a drug for any indication.

References: Since the review is concerned mainly with effects in humans, many of the cited animal studies are not key references. Key works in the field such as the papers by the late Malcolm Lader (e.g.  doi: 10.1111/j.1360-0443.2011.03563.x.) are missing, and must be cited. The references in Spanish language are not helpful for the international reader, and more English references would be desirable.

Reviewer 3 Report

This is an important review summarizing benzodiazepine pharmacology, toxicity, and benefits and problems associated with their use and misuse worldwide. The review is well-written and covers several important topics related to benzodiazepine use.

I enjoyed reading the review, and the authors do a great job at emphasizing the therapeutic utility of benzodiazepines while also discussing their potential dangers. The first paragraph of item 7.1 is great. Given the current climate, it might be worth mentioning as well the major increase in benzodiazepine use and prescription during the COVID-19 pandemic (see the 2020 America’s State Of Mind report by Express Scripts for actual numbers – “Prescriptions for anti-anxiety medications increased 34.1% from mid-February to mid-March”).

The authors should provide a more careful description of benzodiazepine pharmacology in general. For example, the authors need to make clear that benzodiazepines exert their pharmacological effects specifically by binding to γ-aminobutyric acid type A (GABAA) receptors, with their major targets being GABAA receptors containing α1, α2, α3 and α5 subunits (α1GABAA, α2GABAA, α3GABAA, and α5GABAA, respectively; for reviews, see Engin et al. 2018 Trends Pharmacol Sci 39:710-732). Although other alpha subunits exist, benzodiazepines are known to only bind to GABAA receptors containing these specific alpha subunits.

The authors should also make it clear that each receptor subtype seems to mediate the different behavioral effects of benzodiazepines, but that the alpha subunit determines these effects. Also, the role of each of the alpha subunits in the effects of benzodiazepines is not completely elucidated. There seem to be differences in results obtained with rodents (see Engin et al. 2018 Trends Pharmacol Sci 39:710-732) vs nonhuman primates (see Meng et al., 2020 J Psychopharmacol. 34(3):348-357). Generally, it seems like alpha2 subunits mediate the anxiety-reducing effects of benzodiazepines, while alpha1 and alpha3 subunits mediate their abuse-related effects (which should also be mentioned in the review). However, this is not entirely clear given the lack of availability of alpha2-selective compounds to further test these hypotheses, so the authors should be careful about their wording. Similarly, please reword the statement on item 12 that “the α1 sub-unit is the responsible for the dependence effect” (this hasn’t been proven in primates).

When discussing benzodiazepine pharmacology, it would also be important to discuss how different benzodiazepine-type compounds may have selective affinity vs selective efficacy for different GABAA receptor subtypes.

Another important topic that the authors might consider mentioning under Topic 7 is the increasing use of benzodiazepines in combination with opioids, particularly in the United States, where opioid addiction is currently considered an epidemic. Nearly 30% of opioid overdose deaths are associated with the use of a benzodiazepine, which indicates that benzodiazepines can potentiate opioid-induced respiratory depression (some studies have been published indicating that, e.g. Baillargeon et al., Ann Am Thorac Soc. 2019 Oct;16(10):1245-1251).

I believe “Benzodiazepine dependence” also deserves its own subtopic (perhaps under Topic 7). It would be informative to mention in such subtopic that the currently available treatment options for benzodiazepine dependence (dose tapering, gabapentin, phenobarbital) are ineffective and, in the case of barbiturates, dangerous (and defeat the purpose of benzodiazepines in the first place, as they were proposed as a replacement for the toxic and dangerous barbiturates). Also, benzodiazepine withdrawal is known to be extremely aversive in humans, which makes it difficult for individuals to stop benzodiazepine use and abuse.

Under the topic “Who prescribes benzodiazepines”, the authors should also discuss how long benzodiazepines are generally prescribed for vs how long they should be prescribed for.

On Item 12, the authors should cite the extensive work by Dr. James Rowlett on benzodiazepine-type compounds with different subtype selectivity.

Minor comments:

The manuscript is really well-written, but some translation mistakes are still present (e.g. publicated vs published, victims given positive vs victims tested positive). Please check the text thoroughly.

Reviewer 4 Report

While the purpose of the review and the conclusions are absolutely worth of notice, in my opinion, the body does not add, or gathers, any new information on the use and misuse of benzodiazepines. I appreciate the pharmacological overview on the subject, but I think that the examples cited to highlight the improper use of benzodiazepines are closer to a forensic and social view, rather than a scientific one, and not always accurate. I recommend major revision of the manuscript before resubmitting it.